# Amelioration of Biogas Production from Waste-Activated Sludge through Surfactant-Coupled Mechanical Disintegration

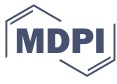

**Vijetha Valsa** [1,†], **Geethu Krishnan S** [1,†], **Rashmi Gondi** [1], **Preethi Muthu** [2], **Kavitha Sankarapandian** [3], **Gopalakrishnan Kumar** [4,5], **Poornachandar Gugulothu** [6] and **Rajesh Banu Jeyakumar** [6,*]

[1]  Department of Life Sciences, Central University of Tamil Nadu, Thiruvarur 610005, Tamil Nadu, India
[2]  Department of Physics, Regional Campus, Anna University, Tirunelveli 627007, Tamil Nadu, India
[3]  Department of Civil Engineering, Anna University Regional Campus, Tirunelveli 627007, Tamil Nadu, India
[4]  Institute of Chemistry, Bioscience and Environmental Engineering, Faculty of Science and Technology, University of Stavanger, 4036 Stavanger, Norway
[5]  School of Civil and Environmental Engineering, Yonsei University, Seoul 03722, Republic of Korea
[6]  Department of Biotechnology, Central University of Tamil Nadu, Thiruvarur 610005, Tamil Nadu, India
*  Correspondence: rajeshbanu@cutn.ac.in
†  These authors contributed equally to this work.

**Abstract:** The current study intended to improve the disintegration potential of paper mill sludge through alkyl polyglycoside-coupled disperser disintegration. The sludge biomass was fed to the disperser disintegration and a maximum solubilization of 6% was attained at the specific energy input of 4729.24 kJ/kg TS. Solubilization was further enhanced by coupling the optimum disperser condition with varying dosage of alkyl polyglycoside. The maximum solubilization of 11% and suspended solid (SS) reduction of 8.42% were achieved at the disperser rpm, time, and surfactant dosage of 12,000, 30 min, and 12 μL. The alkyl polyglycoside-coupled disperser disintegration showed a higher biogas production of 125.1 mL/gCOD, compared to the disperser-alone disintegration (70.1 mL/gCOD) and control (36.1 mL/gCOD).

**Keywords:** solubilization; paper mill sludge; suspended solids; disperser; specific energy input

## 1. Introduction

The pulp and paper mill industry is the major industrial sector that governs a significant part of economic growth around the world, while it is a water-exhaustive industry that utilizes about 10–100 m$^3$ of water. Moreover, it produces 0.2 to 0.6 wet tons of sludge per ton of paper produced [1]. The produced paper mill waste-activated sludge (PMWAS) from secondary clarifiers is potentially contaminating and needed to be treated before disposal to reduce the impact on the environment [2]. PMWAS is primarily composed of various microorganisms that can degrade organic pollutants in wastewater by producing large amounts of hydrolytic enzymes [3]. To reduce the effect of secondary pollutants on the ecosystem, waste-activated sludge should be treated before its disposal [4]. Due to environmental risks and budgetary constraints, the traditional sludge disposal techniques, including incineration, ocean dumping, land application, and composting, are highly unreliable and, thus, a proper treatment method is needed [5].

Anaerobic digestion (AD) is one such method that provides an efficient and sustainable technology for treating PMWAS [4]. AD is a four-step complex process involving hydrolysis, acidogenesis, acetogenesis, and methanogenesis [5]. As a result of this process, the volume of sludge is reduced, thus leading to sludge stabilization, eradication of pathogenic organisms, and production of bioenergy [6]. Generally, microbes and enzymes are the integral part in order to depict the biological features of the sludge. Protease and hydrolase enzymes while undergoing hydrolysis, acetate kinase during acidogenesis, and coenzyme F$_{420}$ during methanogenesis are the major enzyme systems which contribute

during AD. Among others, hydrolase composition is more tedious and, thus, this enzyme system is necessary to promote the disintegration of organic matters, which leads to a higher degradation efficiency. Thus, hydrolysis is found to be a rate-limiting step that hinders the biodegradation of organic matters and, thus, reduces bioenergy generation [4,7]. To enhance this step, a proper pretreatment of WAS is necessary, which augments the solubilization of organic contents and biogas production [7]. The pretreatment methods that are currently in application are mechanical, thermal, chemical, and biological [8]. Among these techniques, mechanical handling via dispersion has more potential in improving the solubilization rate through cell wall disruption and leakage of organic content into the liquid phase, thereby augmenting biogas generation during the digestion process [5]. Disperser works on the rotor–stator principle. The sludge biomass is dispersed through an axial drive in the disperser head, and this dispersed biomass is pulled toward the slit of the disperser using an axial drive. Thus, a shear force is generated due to the operation of circumferential speed and the gap between the stator and the rotor, thereby leading to the efficient disintegration of biomass [9]. Apart from its advantages, the major drawback is that this process is energy intensive and, thus, needs coupling with pretreatment methods to overcome this issue.

The disperser-coupled surfactant pretreatment is one of the efficient methods to alleviate the organics in a liquid medium through the fragmentation of sludge biomass [10]. Alkyl polyglycoside (APG) is a supreme class of sugar-based non-ionic surfactants, which contains a saccharide unit and a hydrophobic alkyl chain [11], is commonly synthesized from fatty alcohol and glucose, and, therefore, is easily biodegradable while possessing a lower toxicity [12]. It is used to accelerate the short-chain fatty acid synthesis and the decomposition of organic matters in WAS [13]. It can speed up the solubilization process, thereby enhancing hydrolysis during anaerobic digestion [14]. APG can reduce the immobilization of the floc matrix, unleash the trapped enzyme, and stimulate the release of organic matters from the floc matrix [15]. Therefore, disperser-mediated APG pretreatment enhances the speed and rate of hydrolysis. Commonly, the surfactant alters the structure of microbial cells by detaching the attached site on cell surface to the aqueous phase. APG combines with the hydrophilic protein, weakens the microbial membrane, and causes a disturbance in the cell structure. Moreover, it combines with hydrophobic lipids to cause the liquefaction of cell membrane and impairs the properties that are a major barrier [14]. It furthermore decreases the surface tension and, thus, the coupling of a disperser pretreatment with APG enhances the speed and rate of hydrolysis as well as decreases the energy used up during the disperser pretreatment. Hitherto, no studies have been developed by coupling a disperser and a surfactant for the disintegration of sludge biomass.

In this study, APG was coupled with a disperser pretreatment to intensify the biomethane production potential. The detailed methodology of the present study is illustrated in Figure 1. The major aims of this study are (1) to assess the disintegration efficiency of the disperser pretreatment, (2) to assess the influence of the APG-coupled disperser pretreatment on sludge biomass disintegration using soluble organic release, and (3) to assess the impact of the combined APG and disperser pretreatment on biogas production.

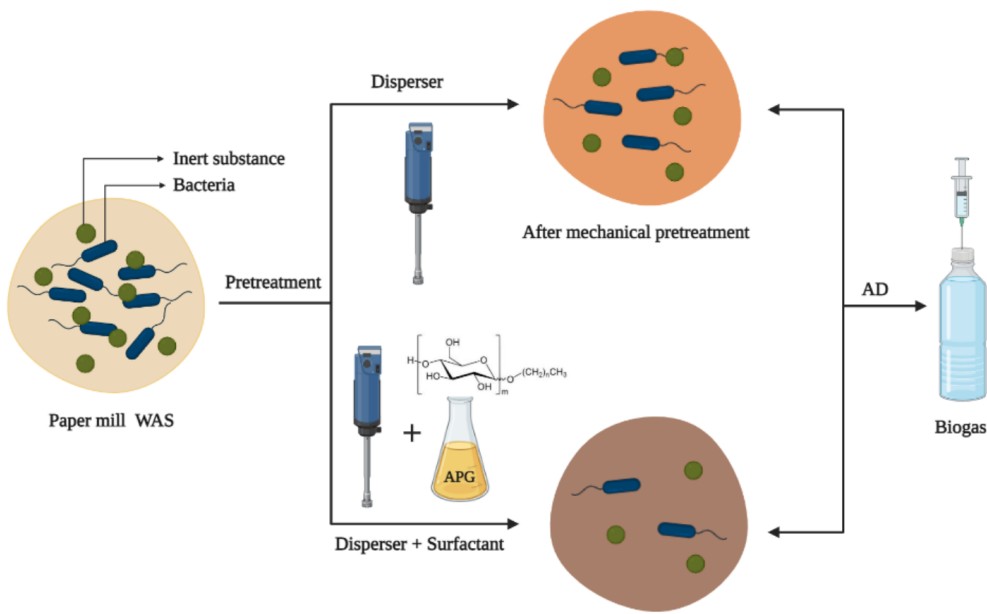

**Figure 1.** Detailed methodology of the present study.

## 2. Materials and Methods

In the present study, an investigation of the effect of APG-D on mechanical disintegration was carried out. An illustration of this study is depicted in Figure 1.

### 2.1. Sample Collection and Characterization

The pulp and paper mill sludge (PPMS) was collected from a pulp and paper mill situated in Tirunelveli, Tamil Nadu, India. The gathered sample was stored at 4 °C, and the initial characterization was determined as follows: mixed liquor suspended solids (MLSS)—7000 mg/L, soluble chemical oxygen demand (SCOD)—100 mg/L, total chemical oxygen demand (TCOD)—3200 mg/L, protein—6 mg/L, and carbohydrate—4 mg/L.

### 2.2. Disperser Disintegration

Disperser disintegration was carried out using an IKAT25 Digital Ultra Turrax disperser, with a cutting rod of S25N. A total of 500 mL of PMS was poured into a 1 L beaker, and the experiment was performed by varying the disperser rpm from 6000 to 20,000 at a pH of 6.8 and a temperature of 35 °C. The sample was collected at regular intervals for a total period of 60 min. The collected samples were centrifuged at 12,000 rpm and the supernatant was examined for organic release. The disperser specific energy was calculated based on the study by Yukesh Kannah et al. [16]

$$\text{Specific Energy Input (kJ/kg TS)} = \frac{P \times t}{V \times TS}$$

where P is the disperser power in kW; t is the time in sec; V is the volume of sample in L; and TS is the total solid concentration in mg/L.

### 2.3. Surfactant-Coupled Disperser Disintegration

In this experiment, the disperser was coupled with the surfactant alkyl polyglycoside (APG). A total of 500 mL of PMS was poured into a 1 L beaker and added with varying APG dosage from 2 to 20 μL at the optimal disperser rpm. The sample was collected at regular intervals and centrifuged at 12,000 rpm, and the supernatant was collected for further analysis.

### 2.4. Anaerobic Biodegradability Assay

An anaerobic biodegradability assay was performed as described in a study by Kavitha et al. [5]. The experiment was conducted in a 300 mL reactor with an inoculum-to-substrate ratio of 3:1. Bovine rumen fluid was used as an inoculum since it has a higher amount of anaerobic bacteria and, thus, has a higher efficiency in degrading organics. The substrate used for the methane generation was an untreated sludge sample (C), a disperser-disintegrated sludge sample (DD), or an APG-coupled disintegrated sludge sample (APG-D). After maintaining the anaerobic condition in the reactor by purging it with nitrogen gas, the reactor bottles were sealed with a rubber septum and covered with aluminum foil. The septum was injected with a syringe and kept in an orbital shaker at 150 rpm. The produced biogas was estimated based on the volume of the syringe displaced.

### 2.5. Analytical Methods

Soluble organic release, total organic release, and suspended solids were determined as per APHA standards. The protein estimation was carried out using the Lowry's method, and the carbohydrate quantification was performed using the anthrone–sulphuric acid method [17].

## 3. Results and Discussion

### 3.1. Disperser Disintegration

The efficacy of disperser disintegration was investigated by examining the variation in soluble organic release (SOR) and suspended solid reduction (SS).

### 3.1.1. Soluble Organic Release

Disperser disintegration was performed by varying the disperser rpm and time to gauge the SOR. Disperser-assisted cell wall cleavage works on three major mechanisms: firstly, via cavitation; secondly, via the generation of hydroxyl radical; and, thirdly, via the shear force generation [5]. The first mechanism occurs via the generation of microbubbles during dispersion, which leads to cavitation. Secondly, the cavitation generates hydroxyl radicals in liquid phase due to the hemolytic splitting of hydrogen bonds. These generated radicals interact with the cells of the sludge biomass to release the intracellular components to the aqueous phase. In the third mechanism, the shear force produced due to the circumferential speed and space between the rotor and the stator cleaves the cell wall of the sludge biomass and augments the soluble organics [18]. Thus, the combination of these three mechanisms encourages solubilization and solid reduction by loosening the internal region of the biomass and, thus, enhancing the surface area of the biomass structure. Figure 2a depicts the effects of disperser rpm and time on SOR and solubilization. It is evident from the figure that SOR increases with an increase in time till 30 min for all the rpm ranging from 6000 to 20,000. Beyond 30 min, a steady release is noted at all disperser rpm. For example, at 6000 rpm, SOR increases from 100 to 160 mg/L, while increasing the time from 0 to 30 min. With further increase in the time from 45 to 60 min, the release is observed to be 161–163 mg/L, which shows a steady release. Likewise, a similar trend of release was obtained at all disperser rpm. This trend of release is similar to Sethupathy et al. [18], where the surfactant-coupled disperser pretreatment method was adopted to pretreat the sludge. Thus, the results show that the maximum SOR is assumed to be released at 30 min and is considered to be optimal.

Apart from the disintegration time, the disperser rpm serves as an important factor to be optimized for evaluating the efficiency of SOR. From 6000 to 12,000 rpm, the release was observed to be 160–195 mg/L. This might be due to the efficient breakdown of biomass cells and the release of intracellular components to the soluble phase through the action of cavitation, shear force, hydroxyl radical action, and thermal effect. Beyond 12,000 rpm, there was no notable rise in SOR. For instance, at 14,000 rpm, the SOR was found to be 196 mg/L. While comparing this with 12,000 rpm, a mere release was observed, and, thus, further increase in rpm beyond 12,000 leads to the consumption of energy rather than SOR.

A statistical one-way analysis was performed to assess the variation in SOR due to the variation in disperser rpm from 6000 to 20,000 rpm. For the disperser rpm from 6000 to 10,000, the *p*-values were calculated to be 0.50, showing that these values are not significant. While increasing the disperser rpm from 10,000 to 12,000, the *p*-value was below 0.05 and noted to be 0.02. Again, while comparing the mean data of SOR for disperser rpm varying from 12,000 to 2000, an insignificant different was found with the *p*-value being much greater than 0.05 and noted to be 0.98. The disperser rpm is comparable with the study by Kavitha et al. [5], where they obtained the maximum SOR at a disperser rpm of 12,000. In contrast, the study conducted by Kumar et al. [10] shows the optimum rpm to be 10,000 for achieving 1250 mg/L of SOR in macroalgal biomass. Thus, the optimum rpm at which SOR is higher is considered to be 12,000.

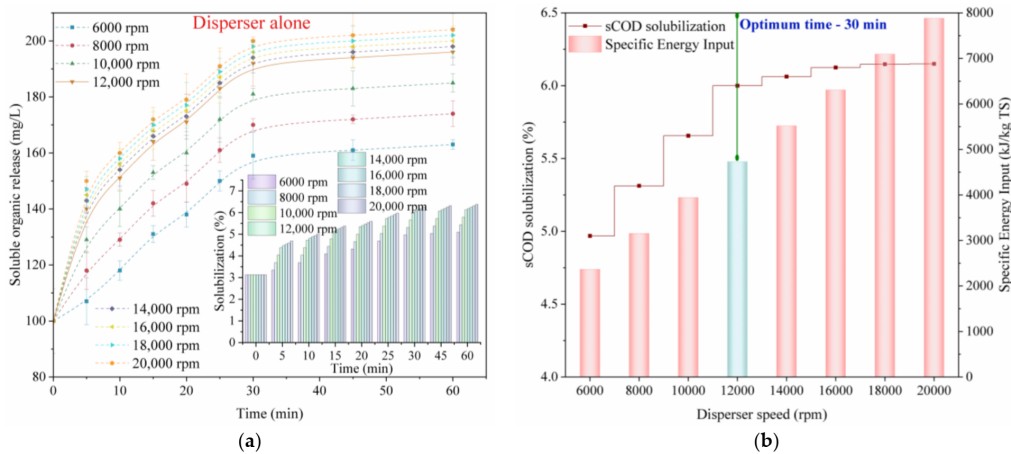

**Figure 2.** (**a**) Effect of disperser pretreatment on soluble organic release. (**b**) Impact of disperser rpm on specific energy and COD solubilization.

Figure 2b shows the effect of disperser rpm on specific energy input and solubilization of the sludge biomass at the optimum disintegration time of 30 min. It can be observed from the figure that solubilization follows two phases, an incremental phase from 6000 to 12,000 rpm and a steady phase from 14,000 to 20,000 rpm. In the incremental phase, solubilization is observed to be increasing from 4.9 to 6%, while increasing the disperser rpm from 6000 to 12,000. During this phase, the maximum specific energy input of 4729.26 kJ/kg TS is utilized to achieve the maximum solubilization. In comparison to the present study, the work by Sethupathy and Sivashanmugam [4] achieved a maximum solubilization of 19% at a specific energy input of 8547 kJ/kg TS. In the steady phase, solubilization is not appreciable (6.06–6.15%) compared to the incremental phase, while the specific energy spent is higher (7882.1 kJ/kg TS). Based on the aforementioned facts, the disperser rpm of 12,000 and specific energy input of 4729.26 kJ/kg TS are considered to be optimum.

### 3.1.2. Suspended Solid Reduction

Suspended solid (SS) is an index to evaluate the stability of sludge by examining the variation in solid changes during disperser disintegration. Figure 3 depicts the impact of the disperser pretreatment time and rpm on SS reduction. The trend shown on the SS reduction graph is analogous to that of solubilization. From the figure, it is noticed that the SS reduction also follows two phases, namely a rapid reduction phase from 0 to 30 min and a slow stabilization phase from 45 to 60 min at all disperser rpm. For instance, at 6000 rpm, the rapid reduction phase was found to be between 0 and 30 min with a SS reduction percentage between 0 and 3.85%, whereas the stabilization phase was found to be from 45 to 60 min with a percentage of 3.85–4%. A similar trend at all the rpm shows that 30 min is optimum for SS reduction

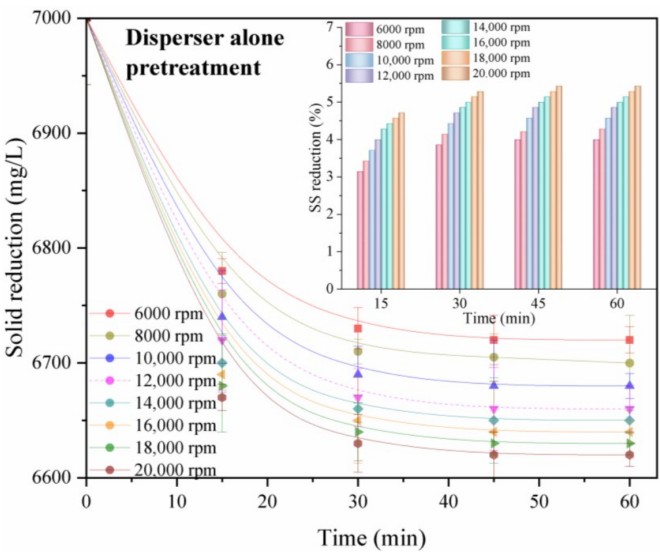

**Figure 3.** Effect of disperser pretreatment on suspended solid reduction.

A step-by-step increment in SS reduction was noted when the disperser rpm increased from 6000 to 12,000 rpm with a SS reduction percentage of 4.96–4.71%. This shows that the structural breakdown of biomass is due to the mechanical shear force caused by the disperser [4]. In contrast, an insignificant SS reduction was observed between 14,000 and 20,000 rpm, with the SS reduction efficiency being between 5 and 5.42%. Thus, the optimum reduction efficiency was achieved at 12,000 rpm.

### 3.2. Alkyl Polyglycoside-Coupled Disperser Disintegration

The combined action of surfactant, alkyl polyglycoside, and disperser was performed by varying the dosage of alkyl polyglycoside from 2 to 20 µL at the optimum disperser rpm of 12,000. The efficiency of this coupled pretreatment was assessed by examining SOR and SS reduction.

#### 3.2.1. Soluble Organic Release

Alkyl polyglycoside is a non-ionic surfactant that reduces the surface energy of the medium. Furthermore, the reduction in surface energy leads to a reduction in surface tension and an enhancement in the surface area of the medium, thus causing an easier penetration to the cell wall of biomass; alkyl polyglycoside gets adsorbed on the cell wall, causing lysis and discharge of intracellular components to the soluble phase [9]. The crushing impact of disperser disintegration enhances the surface energy in the medium and, thus, leads to the formation of clusters of biomass. This in turn causes a reduction in disperser disintegration efficiency. The surfactant-coupled pretreatment method helps in reducing the surface energy and prevents cluster formation [19]. Moreover, it supports the hydromechanical shear force and cavitation process in the medium and, thus, hastens the release of organic compounds to the soluble phase.

Figure 4a,b illustrate the effect of surfactant concentration at the optimum disperser rpm on SOR and sCOD solubilization. The SOR and sCOD solubilization patterns are distinguished into double phases, such as a quicker incremental and a steady phase. As the AGP dosage increase from 2 to 12 µL, a quicker increment in SOR and solubilization occurs. During this dosage of surfactant, SOR increases from 308 to 352 mg/L, with a solubilization of 9.62–11%, respectively. Moreover, the solubilization in combined disintegration (11%) is higher than that of the dispersion pretreatment (6%) alone, as evidenced in Figures 2 and 4. This increment might be due to the addition of the surfactant, where the cleavage of sludge components takes place due to the existence of electrostatic interaction between the enzymes and organic matter. Moreover, APG reduces the surface tension and, thus, aids in enhancing the disintegration efficiency of the sludge biomass by making it more vulnerable

to the effect of mechanical dispersion [20]. The synergic effect of AGP with disperser action on the sludge biomass ruptures the cell wall and releases the intracellular components to the aqueous phase. The obtained SOR is analogous to the study by Kumar et al. [21] and Tamilarasan et al. [22], where they achieved 1603 mg/L and 1400 mg/L of SOR during a combined electrolysis and sonic pretreatment of microalgal biomass and a combined surfactant and disperser pretreatment of macroalgal biomass.

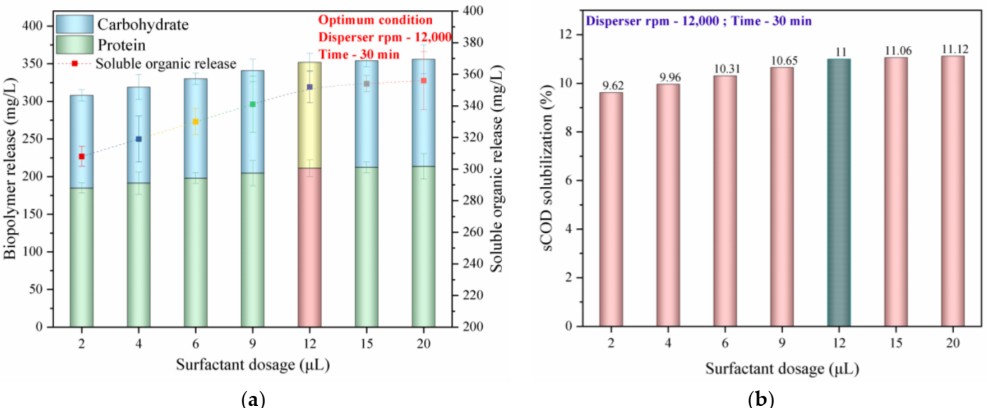

**Figure 4.** Effect of the surfactant-coupled disperser pretreatment on (**a**) soluble organic release and biopolymer, and on (**b**) solubilization.

The lower SOR in the present study than in the other studies might be due to the nature of the biomass and the pretreatment condition. The steady phase occurred when the surfactant dosage increased from 15 to 20 µL, where SOR and solubilization were very scanty, with SOR of 354–356 mg/L and solubilization of 11.06–11.12%, respectively. The increment in surfactant dosage beyond 12 µL can simply cause an increase in cost incurred for chemicals rather than the solubilization of sludge biomass. Based on the one-way ANOVA, an APG dosage between 9 and 12 µL shows significant result with a *p*-value less than 0.05, whereas dosages from 2 to 9 µL and from 12 to 20 µL show the *p*-values of 0.86 and 0.95. Thus, these results show that an AGP dosage of 12 µL is suitable during the combined disintegration of sludge biomass.

Sludge biomass is rich in protein and carbohydrates and is, thus, considered to be the principal component. The soluble biomolecule acts as a suitable substrate for methanogens during anaerobic digestion and, thus, improves biogas production [4]. Figure 4a depicts the concentration profile of biopolymers during combined disintegration. Similar to solubilization, a phase separation occurs during biopolymer release, with the first incremental phase from 2 to 12 µL and the second stabilizing phase from 15 to 20 µL. This trend of biopolymer release is analogous to the results reported in the study by Kumar et al. [10], in which Tween 80-mediated disperser disintegration pretreatment was employed to treat macroalgal biomass *Ulva reticulata*. In the first phase, a linear increment in protein and carbohydrate was seen with a release from 184.8 to 211.2 mg/L and from 123.2 to 140.8 mg/L, respectively, when the AGP dosage increased from 2 to 12 µL. This increment is due to the efficient effect of the surfactant-coupled disperser disintegration of sludge biomass, thus leading to the increase in soluble protein and carbohydrate due to microbial cell lysis. With further increase in AGP dosage from 15 to 20 µL, the release becomes almost stable with a release of 212.4–213.6 mg/L (protein) and 141.6–142.4 mg/L (carbohydrate), respectively. A similar result was obtained by Xiao et al. [14] during a microwave-coupled APG pretreatment. Thus, the optimum APG dosage of 12 µL is efficient for the APG-coupled disperser disintegration of sludge biomass.

### 3.2.2. SS Reduction

Figure 5 depicts the effect of the surfactant-coupled disperser pretreatment on suspended solid reduction. The APG dosage was varied from 2 to 20 µL at the optimum

disperser condition of 12,000 rpm and 30 min. With an increase in the concentration of the surfactant from 2 μL to 12 μL, there is a change in solids, with the SS reduction from 6480 to 6400 mg/L and the reduction efficiency between 7.43 and 8.43%. This reduction may be due to the synergetic effect of the combined pretreatment, which leads to a compression in the sludge surface, causing the disintegration of the biomass and, thus, resulting in the release of intracellular components to the soluble phase. Beyond 12 μL, there is no remarkable reduction in SS and the reduction efficiency is found to be 8.61–8.64%, when the APG dosage increases from 15 μL to 20 μL. A similar trend was observed by Ushani et al. [23] when dioctyl sodium sulphosuccinate was used as the surfactant and coupled with a sonic pretreatment for the disintegration of waste-activated sludge.

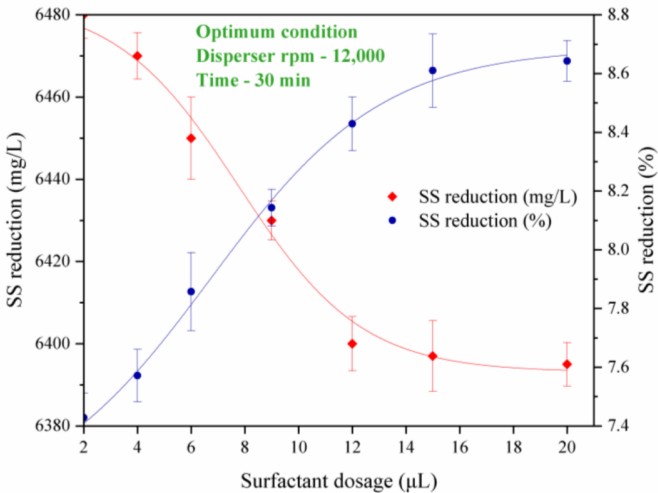

**Figure 5.** Effect of the surfactant-coupled disperser pretreatment on suspended solid reduction.

### 3.3. Anaerobic Biodegradability Assay

Figure 6 depicts the biogas production for the control (C) sample, the disperser-alone (DD) pretreated sample, and the alkyl polyglycoside-induced disperser (APG-D) pretreated sludge sample. Biogas production is lower at the beginning and steady at the end of the digestion, similar to previous findings [24]. The trend of the graph is distinguished into three phases—a lag phase, a log phase, and a stationary phase [10]. The biogas production on the 6th day is noted to be 2, 17, and 39 mL/gCOD in the C, DD, and APG-D samples. This lower biogas production is due to the adaptation of anaerobic bacteria to their new habitat [23]. Beyond the 6th day, biogas production increases in all the samples till the 15th day. In the case of the control, the maximum biogas production of 36.1 mL/gCOD was achieved on the 15th day. This lower release might be due to the presence of an extracellular layer and tough cell wall, which restricts the activity of methanogens on organic matters. In the DD and APG-D samples, biogas production was found to be 70.1 and 125.1 mL/gCOD, respectively, on the 15th day. Elevated gas generation in the APG-D pretreated sample might be due to the reduction in surface tension of the particles by the surfactant and, subsequently, the disperser pretreatment enhances the surface area for anaerobic microbial action, thus showing a rapid hydrolysis of the sludge biomass. Thus, the synergic action between the surfactant and the sonic pretreatment enhances biogas production [5]. The upsurge in biogas generation is analogous to the study conducted by [23,25]. As expected, biogas production was found to be low in the control compared to the pretreated samples. This might be due to the rate-limiting hydrolysis step during AD and the reduced availability of substrates to microorganisms for biogas production [10]. In a study by Zhao et al. [13], the presence of APG shows an inhibitory effect on methanogenic bacteria by rupturing the cell membrane of methanogens. In the present study, the coupling of a lower dosage of APG with a disperser enhances biogas production. Beyond 15 days, the release becomes stable due to the complete

exploitation of substrate and the beginning of endogenous respiration of microbes. The $R^2$ value is in the range between 0.994 and 0.997 in all the samples; thus, the logistic model shows goodness of fit with the experimental values These results indicate that the AGP-D pretreated sample shows better biogas production compared to the DD and the C samples. A limitation of the present study is the interference of recalcitrant lignocellulosic matters present in the substrate with the disperser pretreatment, which affects disintegration efficiency. Although the APG is efficient in dissolving sludge and the coupling of APG with a disperser is efficient in disintegrating organic matters to the soluble phase, the disintegration efficiency is low. It can be enhanced by the co-digestion of PMS with other waste feedstock, which eventually increases the organic content and, thus, helps in enhancing disintegration and solubilization [26]. The productivity in AD can be enhanced by adopting a two-stage digester.

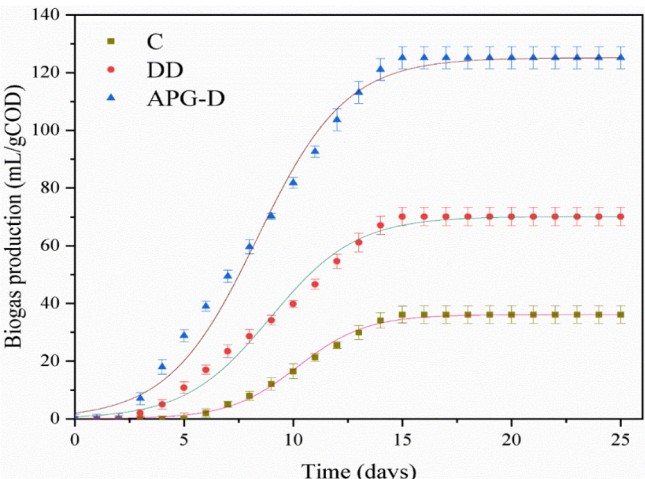

**Figure 6.** Biogas production from the control sample, disperser-alone sample, and surfactant-coupled disperser pretreated sample.

## 4. Conclusions

Surfactant-coupled disperser disintegration of sludge is considered to be a viable feedstock for biogas production. It helps in enhancing the biodegradation of organics by anaerobes and, thus, improves the biogas production potential. The coupling of pretreatment helps in improving efficiency by reducing cost since APG reduces surface energy and prevents cluster formation and, thus, helps in enhancing the efficiency of disperser disintegration. The disperser rpm of 12,000, APG dosage of 12 µL, and pretreatment time of 30 min are optimum for APG-coupled disperser pretreatment, with a maximum solubilization of 11%. The availability of soluble organics is efficiently consumed by anaerobic microbes to produce biogas of 125.1 mL/gCOD. Thus, APG-coupled disperser pretreatment is an efficient disintegration method than disperser-alone pretreatment. In full-scale application, the complete process takes place in continuous mode; thus, the APG-D must be performed continuously and it is necessary to assess the feasible operating conditions. Since a lower APG dosage is efficient for sludge disintegration, foaming is highly avoided.

**Author Contributions:** V.V. and G.K.S.: Wet chemical analysis and writing; R.G. and P.M.: Analysis of data and writing; K.S.: Conceptualization; G.K.: Resources; P.G.: Resources; R.B.J.: Supervision, data validation, and project lead. All authors have read and agreed to the published version of the manuscript.

**Funding:** This work is supported by Department of Biotechnology, India under its initiative Mission innovation Challenge Scheme (IC4). The grant from the project entitled "A novel integrated biorefinery for conversion of lignocellulosic agro waste into value added products and bioenergy" (BT/PR31054/PBD/26/763/2019) is utilized for this study.

**Institutional Review Board Statement:** Not applicable.

**Informed Consent Statement:** Not applicable.

**Data Availability Statement:** Not applicable.

**Conflicts of Interest:** The authors declare no conflict of interest.

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
