# Peer review of "Amelioration of Biogas Production from Waste-Activated Sludge through Surfactant-Coupled Mechanical Disintegration"

_fermentation, doi:10.3390/fermentation9010057_

Round 1

Reviewer 1 Report

Authors have conducted an interesting research and the results presented are expected to contribute to the field of methane production. The comments appended below should help improve the work further:

1 Please add a note for the abbreviations in Table 1.
2 The results obtained in present study should be compared with the latest reported research work, not only with the comparison set in this study

3 Latest trends in the field of methane production should be more effectively discussed by referring to the latest reviews published over the last couple of years.
4 Figure 3 is puzzling, the contents of double coordinate titles are the same, just the units are different, is it necessary to have two coordinate axes?

5 Adding more in-depth discussions on the microbial mechanisms involved would be interesting.

6 The limitations of the study should also be explained.

Author Response

Reviewer 1

S. No.

Reviewer / Editor comments

Author`s Response

1.       

 Please add a note for the abbreviations in Table 1.

Authors are thankful for reviewer’s suggestion. The table is removed and the parameters are explained in text with necessary abbreviations.

2.       

The results obtained in present study should be compared with the latest reported research work, not only with the comparison set in this study

Authors are thankful for reviewer’s suggestion. As per suggestion, the obtained results were compared with other studies.

This trend of release was similar to Sethupathy et al., (2020)  where surfactant coupled disperser pretreatment method was adopted to pretreat the sludge

The disperser rpm is comparable with the study by Kavitha et al., (2016), where they obtained the maximum SOR at the disperser rpm of 12000. In contrast, the study conducted by Kumar et al., (2018), shows the optimum rpm as 10000 for achieving 1250 mg/L of SOR in macroalgal biomass.

While comparing the present study with the work by Sethupathy & Sivashanmugam, (2018), they achieved the maximum solubilization of 19%  at the specific energy input of 8547kJ/kg TS

The obtained SOR was analogous to the study of Kumar et al., (2017) and Tamilarasan et al., (2017) where they achieved 1603 mg/L and 1400 mg/L of SOR during combined electrolysis and sonic pretreatment in microalgal biomass and combined surfactant and disperser pretreatment in macroalgal biomass.

This trend of biopolymer release was analogous as reported in the study by Kumar et al., (2018), when Tween 80 mediated disperser disintegration pretreatment was employed in macroalgal biomass Ulva reticulata.

3.       

Latest trends in the field of methane production should be more effectively discussed by referring to the latest reviews published over the last couple of years.

Authors are thankful for reviewer’s suggestion. As per suggestion, the recent trends were discussed.

Though the APG is efficient in dissolving sludge and coupling of APG with disperser is efficient in disintegrating the organic matter to soluble phase, the disintegration efficiency is low. It can be enhanced by the co-digestion of PMS with other waste feedstock, which eventually increases the organic content and thus helps in enhanced disintegration and solubilization (Atelge et al., 2020). The productivity in AD can be enhanced by adopting two stage digester.

4.       

Figure 3 is puzzling, the contents of double coordinate titles are the same, just the units are different, is it necessary to have two coordinate axes?

Author’s are thankful for reviewer’s suggestion. As per suggestion, figure 3 is revised.

5.       

Adding more in-depth discussions on the microbial mechanisms involved would be interesting.

Commonly, the surfactant alters the structure of microbial cells by detaching the attached site in cell surface to aqueous phase. APG combines with the hydrophilic protein and weakens the microbial membrane and causes disturb in cell structure. Moreover it combines with hydrophobic lipids to causes liquefaction of cell membrane and impairs the properties that are a major barrier (Xiao & Shen, 2017).

Generally, the microbes and enzymes are integral part in order to depict the biological features of the sludge. Protease and hydrolase enzymes while undergoing hydrolysis, acetate kinase during acidogenesis and coenzyme F420 during methanogenesis are the major enzyme system which contributes during AD. Among others, the hydrolase composition is more tedious and thus this enzyme system is necessary to promote the disintegration of organic matters, which leads to higher degradation efficiency.

This increment might be due to the addition of surfactant, where cleavage of sludge components takes place due to the existence of electrostatic interaction between the enzymes and organic matter. Moreover, APG reduces the surface tension and thus aids in enhancing disintegration efficiency of sludge biomass, by making them more vulnerable to the effect of mechanical dispersion (Maryam et al., 2021).

In a study by Zhao et al., (2015), the presence of APG shows the inhibitory effect on methanogenic bacteria by rupturing the cell membrane of methanogens. While in present study the coupling of lesser dosage of APG with disperser enhances the biogas production.

6.       

The limitations of the study should also be explained.

Author’s are thankful for reviewer’s suggestion. As per suggestion, limitation of the study is explained

The limitation of present study is the interference of recalcitrant lignocellulosic matters present in substrate with disperser pretreatment and thus affects the disintegration efficiency. Though the APG is efficient in dissolving sludge and coupling of APG with disperser is efficient in disintegrating the organic matter to soluble phase, the disintegration efficiency is low. It can be enhanced by the co-digestion of PMS with other waste feedstock, which eventually increases the organic content and thus helps in enhanced disintegration and solubilization.

Reviewer 2 Report

Paper mill waste-activated sludge (PMWAS) is generated in huge amount worldwide. Anaerobic digestion (AD) can be a viable option for the treatment/stabilization before disposal, and utilization of PMWAS. Mechanical disintegration can be utilizable to enhance the efficiency hydrolysis step of AD. Surfactant dosage, such as APG, in disperser-coupled surfactant pre-treatments can decrease the energy demand of mechanical disintegration with higher efficiency which can be resulted in higher biogas yield. Therefore, the topic of the manuscript could be considered as interesting for the readers. But, in my opinion, the manuscript fermentation-2099946 in present form with present content does not achieve the scientific quality for publishing.

Specific comments:

Please define clearly the novelties of the study.

Please reconsider the sentence in line 82.

Please give the characteristics of PMWAAS in the text or in Table 1 (both unnecessary).

The calculation of energy input of disperser is not provided.

The temperature of disintegration is missing (section 2.2).

How was the APG dosage range determined/selected? Please explain it.

Please reconsider the tithe of Figure 1.

Sample coding is not unified (see description in section 2.4 vs. Table 1 or Figure 6.

Establishments and discussion in line 130-137 need references.

The change of sCOD show a ‘saturation’ tendency, in my opinion (Figure 2.b-trendline is not clear).

The visibility of Figure 2 is very poor (mainly labels, text). Please improve it.

Please unify the size of figures.

Establishments and explanation in line 201-210 need references.

In my opinion, ANOVA should be necessary to investigate the significance of the effects process parameters and make possible their optimisation.

I have not found data/information related to the solely effect of APG dosage (without mechanical disintegration) on SOR and SSred (it should be given if ‘synergetic effects’ are investigated).

Authors concluded that APG –DD increased the biogas yield. But, how affect the APG the operation stability of a ‘real’, industry scale AD reactor (mixing, foaming, etc).

Please give the biomethane yield as well, beside the biogas production.

Have the authors information related to the biogas yield from APG?

Line 296: Please make clear which model is mentioned.

Conclusion section is too superficial.

Reference 21 and 22 are the same. Please check the references in the whole manuscript.

Author Response

Reviewer 2

1.       

Please define clearly the novelties of the study.

Authors are thankful for reviewer’s suggestion. As per suggestion, the novelty of the study is defined.

Commonly, the surfactant alters the structure of microbial cells by detaching the attached site in cell surface to aqueous phase. APG combines with the hydrophilic protein and weakens the microbial membrane and causes disturb in cell structure. Moreover it combines with hydrophobic lipids to causes liquefaction of cell membrane and impairs the properties that are a major barrier (Xiao & Shen, 2017).It furthermore decreases the surface tension and thus, the coupling of disperser pretreatment with APG was made to enhance the speed and rate of hydrolysis as well as to decrease the energy used up during disperser pretreatment. Hitherto, no studies has been developed by coupling disperser and surfactant for disintegration of sludge biomass.

2

Please reconsider the sentence in line 82.

Authors are thankful for reviewer’s suggestion. As per suggestion, the title is changed for figure 1.

The detailed methodology of the present study is illustrated in Fig 1

3

Please give the characteristics of PMWAAS in the text or in Table 1 (both unnecessary).

Authors are thankful for reviewer’s suggestion. As per suggestion, the table is removed and the parameters are explained in text.

4

The calculation of energy input of disperser is not provided.

Authors are thankful for reviewer’s suggestion. As per suggestion, the specific energy calculation is provided.

The disperser specific energy was calculated based on the study by (Yukesh Kannah et al., 2019)

Specific Energy Input (kJ/kg TS) =  

Where, P is the disperser power in kW, t is the time in sec, V is the volume of sample in L and TS is the total solid concentration in mg/L.

5

The temperature of disintegration is missing (section 2.2).

Authors are thankful for reviewer’s suggestion. As per suggestion, the changes has been made.

500 mL of PMS was taken in 1 litre beaker and the experiment was performed by varying the disperser rpm from 6000-20000 at sludge temperature and pH of 35 °C and 6.8.

6

How was the APG dosage range determined/selected? Please explain it.

Authors are thankful for reviewer’s suggestion. In most of the studies 20 μL was considered to be optimum and thus the dosage was varied between 2-20 μL in order to assess its effect on sludge biomass while coupling APG and disperser.

7

Please reconsider the title of Figure 1.

Authors are thankful for reviewer’s suggestion. As per suggestion, the title is changed for figure 1.

The detailed methodology of the present study is illustrated in Fig 1

8

Sample coding is not unified (see description in section 2.4 vs. Table 1 or Figure 6.

Authors are thankful for reviewer’s suggestion. As per suggestion, the coding is unified in table, figure and in manuscript.

9

Establishments and discussion in line 130-137 need references.

Authors are thankful for reviewer’s suggestion. As per suggestion, the references are added.

Firstly, the cavitation, secondly, the generation of hydroxyl radical and thirdly due to the shear force generation (Kavitha et al., 2016). The first mechanism occurs by the generation of microbubbles during dispersion which thereby leads to cavitation. Secondly, the cavitation generates hydroxyl radicals in liquid phase due to the haemolytic splitting of hydrogen bonds. These generated radicals interacts with the cells of the sludge biomass to release the intracellular components to the aqueous phase. In third mechanism, the shear force produced due to circumferential speed and space between rotor and stator cleaves the cell wall of the sludge biomass and augments the soluble organics (Sethupathy et al., 2020).

10

The change of sCOD show a ‘saturation’ tendency, in my opinion (Figure 2.b-trendline is not clear).

Authors are thankful for reviewer’s suggestion. As per suggestion, the image quality is improved.

In figure 2b, beyond 12000 rpm the solubilization gets saturated whereas the specific energy input increases as the rpm increases.

11

The visibility of Figure 2 is very poor (mainly labels, text). Please improve it.

Authors are thankful for reviewer’s suggestion. As per suggestion, the image quality is improved

12

Please unify the size of figures.

Authors are thankful for reviewer’s suggestion. As per suggestion, the size of the figures are unified.

13

Establishments and explanation in line 201-210 need references.

Authors are thankful for reviewer’s suggestion. As per suggestion, the references are added.

Furthermore, the reduction in surface energy leads to the reduction in surface tension and enhancement of surface area of media, thus causing easier penetration to the cell wall of biomass and get adsorbed on cell wall causing lysis and discharge of intracellular components to the soluble phase(Kannah et al., 2017).

The surfactant coupled pretreatment method helps in reducing the surface energy and prevents cluster formation(Banu J et al., 2022)

14

In my opinion, ANOVA should be necessary to investigate the significance of the effects process parameters and make possible their optimisation.

Authors are thankful for reviewer’s suggestion. As per suggestion, ANOVA analysis is performed

A statistical one way analysis was performed to assess the variation in SOR due to the variation in disperser rpm from 6000-20000 rpm. For the disperser rpm from 6000-10000, the p value was calculated to be 0.50, showing that the values are not significant. While increasing the disperser rpm from 10000-12000, the p value was below 0.05 and noted to be 0.02. Again while comparing the mean data of SOR for the disperser rpm varying 12000-2000, the insignificant different was found with the p value much greater than 0.05 and noted to be 0.98

Based on one way ANOVA analysis the APG dosage between 9-12 μL shows significant result less than 0.05 whereas the dosage from 2-9 μL and 12-20 μL shows the p values of 0.86 and 0.95.

15

I have not found data/information related to the solely effect of APG dosage (without mechanical disintegration) on SOR and SS red (it should be given if ‘synergetic effects’ are investigated).

Authors are thankful for reviewer’s suggestion.

In present study we investigated the effect of APG coupled disperser disintegration. We performed the experiment by adding different dosage of surfactant at optimized rpm of disperser. We obtained the synergic effect between the disperser and APG and doesnot consider the sole pretreatment. Thus, the sole APG disintegration was not performed and in future studies we will incorporate the effect of APG alone.

16

Authors concluded that APG –DD increased the biogas yield. But, how affect the APG the operation stability of a ‘real’, industry scale AD reactor (mixing, foaming, etc).

Authors are thankful for reviewer’s suggestion. In In full scale application, the complete process takes place in continuous mode and thus the APG-D must be performed continuously and needed to assess the feasible operating conditions. Since the lower APG dosage is efficient for sludge disintegration, the foaming is highly avoided.

17

Please give the biomethane yield as well, beside the biogas production.

Authors are thankful for reviewer’s suggestion. In present study we were much concerned towards the biogas production alone. The biomethane content in biogas was not analysed in this study. We assure that we will include it in our future studies.

18

Have the author’s information related to the biogas yield from APG?

Authors are thankful for reviewer’s suggestion.

Our work is related to the impact of combined APG and disperser pretreatment on soluble organic release for further biodegradation of soluble organics by anaerobes. At first the disperser rpm was optimized and then at optimized rpm APG dosage was varied in order to assess the disintegration efficiency. In this study the APG alone pretreatment was not taken into account.

19

Line 296: Please make clear which model is mentioned.

Authors are thankful for reviewer’s suggestion. As per suggestion, the model is mentioned in the manuscript.

The R2 value is in the range between 0.994-0.997 in all the samples, thus, the logistic model shows goodness of fit with the experimental values

20

Conclusion section is too superficial.

Authors are thankful for reviewer’s suggestion. As per suggestion, the conclusion is rewritten

Surfactant coupled disperser disintegration of sludge is considered to be the viable feedstock for biogas production. It helps in enhancing the biodegradation of organics by anaerobes and thus improves the biogas production potential. The coupling of pretreatment helps in improving the efficiency by reducing cost since APG reduces the surface energy and prevents the cluster formation, and thus helps in enhancing the efficiency of disperser disintegration. The disperser rpm of 12000, APG dosage of 12 μL and the pretreatment time of 30 min was optimum for APG coupled disperser pretreatment with the maximum solubilization of 11%. The availability of soluble organics were efficiently consumed by anaerobic microbes to produce the biogas of 125.1 mL/gCOD. Thus, APG coupled disperser pretreatment is an efficient disintegration method than disperser alone pretreatment.

21

Reference 21 and 22 are the same. Please check the references in the whole manuscript

Authors are thankful for reviewer’s suggestion. As per suggestion, the similar reference is removed from manuscript. The complete list of references are checked.

Round 2

Reviewer 1 Report

All the comments have been well addressed.

Reviewer 2 Report

The manuscript has an interesting and relevant topic. Authors have revised the manuscript thorougfully according to reviewers comments and suggestions. The overall scientific quality of the manuscript has been improved significantly due to the revision. Amendments, rephrasings, imporvement of the quality of figures, additional data/information, and more detailed discussion of the results made the mnauscript more complete and clear. I agree and accept all modifications made by the authors.